# Distilled Wasserstein Learning for Word Embedding and Topic Modeling

**Hongteng Xu**[1,2]    **Wenlin Wang**[2]    **Wei Liu**[3]    **Lawrence Carin**[2]
[1]Infinia ML, Inc.    [2]Duke University    [3]Tencent AI Lab
`hongteng.xu@infiniaml.com`

## Abstract

We propose a novel Wasserstein method with a distillation mechanism, yielding joint learning of word embeddings and topics. The proposed method is based on the fact that the Euclidean distance between word embeddings may be employed as the underlying distance in the Wasserstein topic model. The word distributions of topics, their optimal transports to the word distributions of documents, and the embeddings of words are learned in a unified framework. When learning the topic model, we leverage a distilled underlying distance matrix to update the topic distributions and smoothly calculate the corresponding optimal transports. Such a strategy provides the updating of word embeddings with robust guidance, improving the algorithmic convergence. As an application, we focus on patient admission records, in which the proposed method embeds the codes of diseases and procedures and learns the topics of admissions, obtaining superior performance on clinically-meaningful disease network construction, mortality prediction as a function of admission codes, and procedure recommendation.

## 1  Introduction

Word embedding and topic modeling play important roles in natural language processing (NLP), as well as other applications with textual and sequential data. Many modern embedding methods [30, 33, 28] assume that words can be represented and predicted by contextual (surrounding) words. Accordingly, the word embeddings are learned to inherit those relationships. Topic modeling methods [8], in contrast, typically represent documents by the *distribution* of words, or other "bag-of-words" techniques [17, 24], ignoring the order and semantic relationships among words. The distinction between how the word order is (or is not) accounted for when learning topics and word embeddings manifests a potential methodological gap or mismatch.

This gap is important when considering clinical-admission analysis, the motivating application of this paper. Patient admissions in hospitals are recorded by the code of international classification of diseases (ICD). For each admission, one may observe a sequence of ICD codes corresponding to certain kinds of diseases and procedures, and each code is treated as a "word." To reveal the characteristics of the admissions and relationships between different diseases/procedures, we seek to model the "topics" of admissions and also learn an embedding for each ICD code. However, while we want embeddings of similar diseases/procedures to be nearby in the embedding space, learning the embedding vectors based on surrounding ICD codes for a given patient admission is less relevant, as there is often a diversity in the observed codes for a given admission, and the code order may hold less meaning. Take the MIMIC-III dataset [25] as an example. The ICD codes in each patient's admission are ranked according to a manually-defined priority, and the adjacent codes are often not clinically-correlated with each other. Therefore, we desire a model that jointly learns topics and word embeddings, and that for both *does not* consider the word (ICD code) order. Interestingly, even in the context of traditional NLP tasks, it has been recognized recently that effective word embeddings may

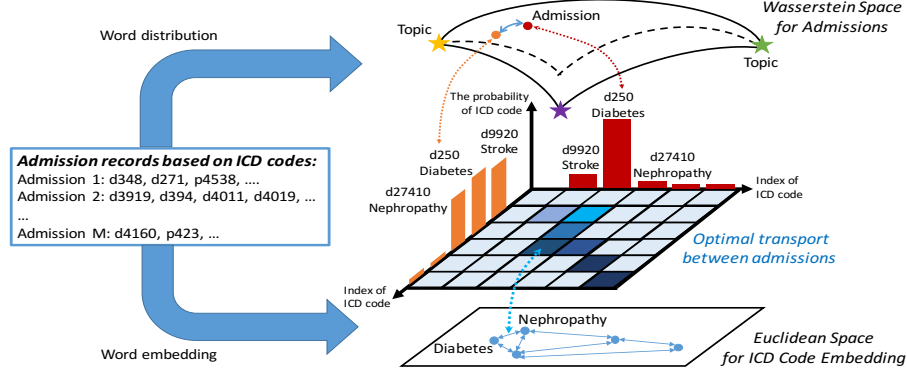

Figure 1: Consider two admissions with mild and severe diabetes, which are represented by two distributions of diseases (associated with ICD codes) in red and orange, respectively. They are two dots in the Wasserstein ambient space, corresponding to two weighted barycenters of Wasserstein topics (the color stars). The optimal transport matrix between these two admissions is built on the distance between disease embeddings in the Euclidean latent space. The large value in the matrix (the dark blue elements) indicates that it is easy to transfer diabetes to its complication like nephropathy, whose embedding is a short distance away (short blue arrows).

be learned without considering word order [37], although that work didn't consider topic modeling or our motivating application.

Although some works have applied word embeddings to represent ICD codes and related clinical data [11, 22], they ignore the fact that the clinical relationships among the diseases/procedures in an admission may not be approximated well by their neighboring relationships in the sequential record. Most existing works either treat word embeddings as auxiliary features for learning topic models [15] or use topics as the labels for supervised embedding [28]. Prior attempts at learning topics and word embeddings jointly [38] have fallen short from the perspective of these two empirical strategies.

We seek to fill the aforementioned gap, while applying the proposed methodology to clinical-admission analysis. As shown in Fig. 1, the proposed method is based on a Wasserstein-distance model, in which ($i$) the Euclidean distance between ICD code embeddings works as the underlying distance (also referred to as the cost) of the Wasserstein distance between the distributions of the codes corresponding to different admissions [26]; ($ii$) the topics are "vertices" of a geometry in the Wasserstein space and the admissions are the "barycenters" of the geometry with different weights [36]. When learning this model, both the embeddings and the topics are inferred jointly. A novel learning strategy based on the idea of model distillation [20, 29] is proposed, improving the convergence and the performance of the learning algorithm.

The proposed method unifies word embedding and topic modeling in a framework of Wasserstein learning. Based on this model, we can calculate the optimal transport between different admissions and explain the transport by the distance of ICD code embeddings. Accordingly, the admissions of patients become more interpretable and predictable. Experimental results show that our approach is superior to previous state-of-the-art methods in various tasks, including predicting admission type, mortality of a given admission, and procedure recommendation.

## 2  A Wasserstein Topic Model Based on Euclidean Word Embeddings

Assume that we have $M$ documents and a corpus with $N$ words, $e.g.$, respectively, admission records and the dictionary of ICD codes. These documents can be represented by $\boldsymbol{Y} = [\boldsymbol{y}_m] \in \mathbb{R}^{N \times M}$, where $\boldsymbol{y}_m \in \Sigma^N$, $m \in \{1, ..., M\}$, is the distribution of the words in the $m$-th document, and $\Sigma^N$ is an $N$-dimensional simplex. These distributions can be represented by some basis ($i.e.$, topics), denoted as $\boldsymbol{B} = [\boldsymbol{b}_k] \in \mathbb{R}^{N \times K}$, where $\boldsymbol{b}_k \in \Sigma^N$ is the $k$-th base distribution. The word embeddings can be formulated as $\boldsymbol{X} = [\boldsymbol{x}_n] \in \mathbb{R}^{D \times N}$, where $\boldsymbol{x}_n$ is the embedding of the $n$-th word, $n \in \{1, ..., N\}$, is obtained by a model, $i.e.$, $\boldsymbol{x}_n = g_\theta(\boldsymbol{w}_n)$ with parameters $\theta$ and predefined representation $\boldsymbol{w}_n$ of the word ($e.g.$, $\boldsymbol{w}_n$ may be a one-hot vector for each word). The distance between two word embeddings is denoted $d_{nn'} = d(\boldsymbol{x}_n, \boldsymbol{x}_{n'})$, and generally it is assumed to be Euclidean. These distances can be formulated as a parametric distance matrix $\boldsymbol{D}_\theta = [d_{nn'}] \in \mathbb{R}^{N \times N}$.

Denote the space of the word distributions as the *ambient space* and that of their embeddings as the *latent space*. We aim to model and learn the topics in the ambient space and the embeddings in the latent space in a unified framework. We show that recent developments in the methods of Wasserstein learning provide an attractive solution to achieve this aim.

## 2.1 Revisiting topic models from a geometric viewpoint

Traditional topic models [8] often decompose the distribution of words conditioned on the observed document into two factors: the distribution of words conditioned on a certain topic, and the distribution of topics conditioned on the document. Mathematically, it corresponds to a low-rank factorization of $\boldsymbol{Y}$, *i.e.*, $\boldsymbol{Y} = \boldsymbol{B}\boldsymbol{\Lambda}$, where $\boldsymbol{B} = [\boldsymbol{b}_k]$ contains the word distributions of different topics and $\boldsymbol{\Lambda} = [\boldsymbol{\lambda}_m] \in \mathbb{R}^{K \times M}$, $\boldsymbol{\lambda}_m = [\lambda_{km}] \in \Sigma^K$, contains the topic distributions of different documents. Given $\boldsymbol{B}$ and $\boldsymbol{\lambda}_m$, $\boldsymbol{y}_m$ can be equivalently written as

$$\boldsymbol{y}_m = \boldsymbol{B}\boldsymbol{\lambda}_m = \arg\min_{\boldsymbol{y} \in \Sigma^N} \sum_{k=1}^{K} \lambda_{km} \|\boldsymbol{b}_k - \boldsymbol{y}\|_2^2, \tag{1}$$

where $\lambda_{km}$ is the probability of topic $k$ given document $m$. From a geometric viewpoint, $\{\boldsymbol{b}_k\}$ in (1) can be viewed as vertices of a geometry, whose "weights" are $\boldsymbol{\lambda}_m$. Then, $\boldsymbol{y}_m$ is the weighted barycenter of the geometry in the Euclidean space.

Following this viewpoint, we can extend (1) to another metric space, *i.e.*,

$$\boldsymbol{y}_m = \arg\min_{\boldsymbol{y} \in \Sigma^N} \sum_{k=1}^{K} \lambda_{km} d^2(\boldsymbol{b}_k, \boldsymbol{y}) = \boldsymbol{y}_{d^2}(\boldsymbol{B}, \boldsymbol{\lambda}_m), \tag{2}$$

where $\boldsymbol{y}_{d^2}(\boldsymbol{B}, \boldsymbol{\lambda}_m)$ is the barycenter of the geometry, with vertices $\boldsymbol{B}$ and weights $\boldsymbol{\lambda}_m$ in the space with metric $d$.

## 2.2 Wasserstein topic model

When the distance $d$ in (2) is the Wasserstein distance, we obtain a Wasserstein topic model, which has a natural and explicit connection with word embeddings. Mathematically, let $(\Omega, d)$ be an arbitrary space with metric $D$ and $P(\Omega)$ be the set of Borel probability measures on $\Omega$, respectively.

**Definition 2.1.** *For $p \in [1, \infty)$ and probability measures $u$ and $v$ in $P(\Omega)$, their p-order Wasserstein distance [40] is $W_p(u, v) = (\inf_{\pi \in \Pi(u,v)} \int_{\Omega \times \Omega} d^p(x, y) d\pi(x, y))^{\frac{1}{p}}$, where $\Pi(u, v)$ is the set of all probability measures on $\Omega \times \Omega$ with $u$ and $v$ as marginals.*

**Definition 2.2.** *The p-order weighted Fréchet mean in the Wasserstein space (or called Wasserstein barycenter) [1] of $K$ measures $\boldsymbol{B} = \{b_1, ..., b_K\}$ in $\mathbb{P} \subset P(\Omega)$ is $q(\boldsymbol{B}, \boldsymbol{\lambda}) = \arg\inf_{q \in \mathbb{P}} \sum_{k=1}^{K} \lambda_k W_p^p(b_k, q)$, where $\boldsymbol{\lambda} = [\lambda_k] \in \Sigma^K$ decides the weights of the measures.*

When $\Omega$ is a discrete state space, *i.e.*, $\{1, ..., N\}$, the Wasserstein distance is also called the optimal transport (OT) distance [36]. More specifically, the Wasserstein distance with $p = 2$ corresponds to the solution to the discretized Monge-Kantorovich problem:

$$W_2^2(\boldsymbol{u}, \boldsymbol{v}; \boldsymbol{D}) := \min_{\boldsymbol{T} \in \Pi(\boldsymbol{u}, \boldsymbol{v})} \text{Tr}(\boldsymbol{T}^\top \boldsymbol{D}), \tag{3}$$

where $\boldsymbol{u}$ and $\boldsymbol{v}$ are two distributions of the discrete states and $\boldsymbol{D} \in \mathbb{R}^{N \times N}$ is the underlying distance matrix, whose element measures the distance between different states. $\Pi(\boldsymbol{u}, \boldsymbol{v}) = \{\boldsymbol{T} | \boldsymbol{T}\boldsymbol{1} = \boldsymbol{u}, \boldsymbol{T}^\top \boldsymbol{1} = \boldsymbol{v}\}$, and $\text{Tr}(\cdot)$ represents the matrix trace. The matrix $\boldsymbol{T}$ is called the *optimal transport* matrix when the minimum in (3) is achieved.

Applying the discrete Wasserstein distance in (3) to (2), we obtain our Wasserstein topic model, *i.e.*,

$$\boldsymbol{y}_{W_2^2}(\boldsymbol{B}, \boldsymbol{\lambda}; \boldsymbol{D}) = \arg\min_{\boldsymbol{y} \in \Sigma^N} \sum_{k=1}^{K} \lambda_k W_2^2(\boldsymbol{b}_k, \boldsymbol{y}; \boldsymbol{D}). \tag{4}$$

In this model, the discrete states correspond to the words in the corpus and the distance between different words can be calculated by the Euclidean distance between their embeddings.

In this manner, we establish the connection between the word embeddings and the topic model: the distance between different topics (and different documents) is achieved by the optimal transport between their word distributions built on the embedding-based underlying distance. For arbitrary

two word embeddings, the more similar they are, the smaller underlying distance we have, and more easily we can achieve transfer between them. In the learning phase (as shown in the following section), we can learn the embeddings and the topic model jointly. This model is especially suitable for clinical admission analysis. As discussed above, we not only care about the clustering structure of admissions (the relative proportion, by which each topic is manifested in an admission), but also want to know the mechanism or the tendency of their transfers in the level of disease. As shown in Fig. 1, using our model, we can calculate the Wasserstein distance between different admissions in the level of disease and obtain the optimal transport from one admission to another explicitly. The hierarchical architecture of our model helps represent each admission by its topics, which are the typical diseases/procedures (ICD codes) appearing in a class of admissions.

## 3 Wasserstein Learning with Model Distillation

Given the word-document matrix $\boldsymbol{Y}$ and a predefined number of topics $K$, we wish to jointly learn the basis $\boldsymbol{B}$, the weight matrix $\boldsymbol{\Lambda}$, and the model $g_\theta$ of word embeddings. This learning problem can be formulated as

$$
\begin{aligned}
&\min_{\boldsymbol{B},\boldsymbol{\Lambda},\theta} \sum_{m=1}^{M} \mathcal{L}(\boldsymbol{y}_m, \ \boldsymbol{y}_{W_2^2}(\boldsymbol{B}, \boldsymbol{\lambda}_m; \boldsymbol{D}_\theta)), \\
&s.t. \quad \boldsymbol{b}_k \in \Sigma^N, \text{ for } k = 1, .., K, \text{ and } \boldsymbol{\lambda}_m \in \Sigma^K, \text{ for } m = 1, .., M.
\end{aligned}
\tag{5}
$$

Here, $\boldsymbol{D}_\theta = [d_{nn'}]$ and the element $d_{nn'} = \|g_\theta(\boldsymbol{w}_n) - g_\theta(\boldsymbol{w}_{n'})\|_2$. The loss function $\mathcal{L}(\cdot, \cdot)$ measures the difference between $\boldsymbol{y}_m$ and its estimation $\boldsymbol{y}_{W_2^2}(\boldsymbol{B}, \boldsymbol{\lambda}_m; \boldsymbol{D}_\theta)$. We can solve this problem based on the idea of alternating optimization. In each iteration we first learn the basis $\boldsymbol{B}$ and the weights $\boldsymbol{\Lambda}$ given the current parameters $\theta$. Then, we learn the new parameters $\theta$ based on updated $\boldsymbol{B}$ and $\boldsymbol{\Lambda}$.

### 3.1 Updating word embeddings to enhance the clustering structure

Suppose that we have obtained updated $\boldsymbol{B}$ and $\boldsymbol{\Lambda}$. Given current $\boldsymbol{D}_\theta$, we denote the optimal transport between document $\boldsymbol{y}_m$ and topic $\boldsymbol{b}_k$ as $\boldsymbol{T}_{km}$. Accordingly, the Wasserstein distance between $\boldsymbol{y}_m$ and $\boldsymbol{b}_k$ is $\text{Tr}(\boldsymbol{T}_{km}^\top \boldsymbol{D}_\theta)$. Recall from the topic model in (4) that each document $\boldsymbol{y}_m$ is represented as the weighted barycenter of $\boldsymbol{B}$ in the Wasserstein space, and the weights $\boldsymbol{\lambda}_m = [\lambda_{km}]$ represent the closeness between the barycenter and different bases (topics). To enhance the clustering structure of the documents, we update $\theta$ by minimizing the Wasserstein distance between the documents and their closest topics. Consequently, the documents belonging to different clusters would be far away from each other. The corresponding objective function is

$$
\sum_{m=1}^{M} \text{Tr}(\boldsymbol{T}_{k_m m}^\top \boldsymbol{D}_\theta) = \text{Tr}(\boldsymbol{T}^\top \boldsymbol{D}_\theta) = \sum_{n,n'} t_{nn'} \|\boldsymbol{x}_{n,\theta} - \boldsymbol{x}_{n',\theta}\|_2^2,
\tag{6}
$$

where $\boldsymbol{T}_{k_m m}$ is the optimal transport between $\boldsymbol{y}_m$ and its closest base $\boldsymbol{b}_{k_m}$. The aggregation of these transports is given by $\boldsymbol{T} = \sum_m \boldsymbol{T}_{k_m m} = [t_{nn'}]$, and $\boldsymbol{X}_\theta = [\boldsymbol{x}_{n,\theta}]$ are the word embeddings. Considering the symmetry of $\boldsymbol{D}_\theta$, we can replace $t_{nn'}$ in (6) with $\frac{t_{nn'} + t_{n'n}}{2}$. The objective function can be further written as $\text{Tr}(\boldsymbol{X}_\theta \boldsymbol{L} \boldsymbol{X}_\theta^\top)$, where $\boldsymbol{L} = \text{diag}(\frac{\boldsymbol{T} + \boldsymbol{T}^\top}{2} \mathbf{1}_N) - \frac{\boldsymbol{T} + \boldsymbol{T}^\top}{2}$ is the Laplacian matrix. To avoid trivial solutions like $\boldsymbol{X}_\theta = \boldsymbol{0}$, we add a smoothness regularizer and update $\theta$ by optimizing the following problem:

$$
\min_\theta \mathcal{E}(\theta) = \min_\theta \text{Tr}(\boldsymbol{X}_\theta \boldsymbol{L} \boldsymbol{X}_\theta^\top) + \beta \|\theta - \theta_c\|_2^2,
\tag{7}
$$

where $\theta_c$ is current parameters and $\beta$ controls the significance of the regularizer. Similar to Laplacian Eigenmaps [6], the aggregated optimal transport $\boldsymbol{T}$ works as the similarity measurement between proposed embeddings. However, instead of requiring the solution of (7) to be the eigenvectors of $\boldsymbol{L}$, we enhance the stability of updating by ensuring that the new $\theta$ is close to the current one.

### 3.2 Updating topic models based on the distilled underlying distance

Given updated word embeddings and the corresponding underlying distance $\boldsymbol{D}_\theta$, we wish to further update the basis $\boldsymbol{B}$ and the weights $\boldsymbol{\Lambda}$. The problem is formulated as a Wasserstein dictionary-learning problem, as proposed in [36]. Following the same strategy as [36], we rewrite $\{\boldsymbol{\lambda}_m\}$ and $\{\boldsymbol{b}_k\}$ as

$$
\lambda_{km}(\boldsymbol{A}) = \frac{\exp(\alpha_{km})}{\sum_{k'} \exp(\alpha_{k'm})}, \quad b_{nk}(\boldsymbol{R}) = \frac{\exp(\gamma_{nk})}{\sum_{n'} \exp(\gamma_{n'k})},
\tag{8}
$$

---

**Algorithm 1** Distilled Wasserstein Learning (DWL) for Joint Word Embedding and Topic Modeling

---

1: **Input:** The distributions of words for documents $\boldsymbol{Y}$. The distillation parameter $\tau$. The number of epochs $I$. Batch size $s$. The weight in Sinkhon distance $\epsilon$. The weight $\beta$ in (7). The learning rate $\rho$.
2: **Output:** The parameters $\theta$, basis $\boldsymbol{B}$, and weights $\boldsymbol{\Lambda}$.
3: Initialize $\theta, \boldsymbol{A}, \boldsymbol{R} \sim \mathcal{N}(0, 1)$, and calculate $\boldsymbol{B}(\boldsymbol{R})$ and $\boldsymbol{\Lambda}(\boldsymbol{A})$ by (8).
4: **For** $i = 1, ..., I$
5:    **For** Each batch of documents
6:       Calculate the Sinkhorn gradient with distillation: $\nabla_{\boldsymbol{B}} \mathcal{L}_\tau|_{\boldsymbol{B}}$ and $\nabla_{\boldsymbol{\Lambda}} \mathcal{L}_\tau|_{\boldsymbol{\Lambda}}$.
7:       $\boldsymbol{R} \leftarrow \boldsymbol{R} - \rho \nabla_{\boldsymbol{B}} \mathcal{L}_\tau|_{\boldsymbol{B}} \nabla_{\boldsymbol{R}} \boldsymbol{B}|_{\boldsymbol{R}}, \quad \boldsymbol{A} \leftarrow \boldsymbol{A} - \rho \nabla_{\boldsymbol{\Lambda}} \mathcal{L}_\tau|_{\boldsymbol{\Lambda}} \nabla_{\boldsymbol{A}} \boldsymbol{\Lambda}|_{\boldsymbol{A}}$.
8:       Calculate $\boldsymbol{B}(\boldsymbol{R})$, $\boldsymbol{\Lambda}(\boldsymbol{A})$ and the gradient of (7) $\nabla_\theta \mathcal{E}(\theta)|_\theta$, then update $\theta \leftarrow \theta - \rho \nabla_\theta \mathcal{E}(\theta)|_\theta$.

---

where $\boldsymbol{A} = [\alpha_{km}]$ and $\boldsymbol{R} = [\gamma_{nk}]$ are new parameters. Based on (8), the normalization of $\{\boldsymbol{\lambda}_m\}$ and $\{\boldsymbol{b}_k\}$ is met naturally, and we can reformulate (5) to an unconstrained optimization problem, *i.e.*,

$$\min_{\boldsymbol{A}, \boldsymbol{R}} \sum_{m=1}^{M} \mathcal{L}(\boldsymbol{y}_m, \ \boldsymbol{y}_{W_2^2}(\boldsymbol{B}(\boldsymbol{R}), \boldsymbol{\lambda}_m(\boldsymbol{A}); \boldsymbol{D}_\theta)). \tag{9}$$

Different from [36], we introduce a model distillation method to improve the convergence of our model. The key idea is that the model with the current underlying distance $\boldsymbol{D}_\theta$ works as a "teacher," while the proposed model with new basis and weights is regarded as a "student." Through $\boldsymbol{D}_\theta$, the teacher provides the student with guidance for its updating. We find that if we use the current underlying distance $\boldsymbol{D}_\theta$ to calculate basis $\boldsymbol{B}$ and weights $\boldsymbol{\Lambda}$, we will encounter a serious "vanishing gradient" problem when solving (7) in the next iteration. Because $\text{Tr}(\boldsymbol{T}_{k_m m}^\top \boldsymbol{D}_\theta)$ in (6) has been optimal under the current underlying distance and new $\boldsymbol{B}$ and $\boldsymbol{\Lambda}$, it is difficult to further update $\boldsymbol{D}_\theta$.

Inspired by recent model distillation methods in [20, 29, 34], we use a smoothed underlying distance matrix to solve the "vanishing gradient" problem when updating $\boldsymbol{B}$ and $\boldsymbol{\Lambda}$. In particular, the $\boldsymbol{y}_{W_2^2}(\boldsymbol{B}(\boldsymbol{R}), \boldsymbol{\lambda}_m(\boldsymbol{A}); \boldsymbol{D}_\theta)$ in (9) is replaced by a Sinkhorn distance with the smoothed underlying distance, *i.e.*, $\boldsymbol{y}_{S_\epsilon}(\boldsymbol{B}(\boldsymbol{R}), \boldsymbol{\lambda}_m(\boldsymbol{A}); \boldsymbol{D}_\theta^\tau)$, where $(\cdot)^\tau, 0 < \tau < 1$, is an element-wise power function of a matrix. The Sinkhorn distance $S_\epsilon$ is defined as

$$S_\epsilon(\boldsymbol{u}, \boldsymbol{v}; \boldsymbol{D}) = \min_{\boldsymbol{T} \in \Pi(\boldsymbol{u}, \boldsymbol{v})} \text{Tr}(\boldsymbol{T}^\top \boldsymbol{D}) + \epsilon \text{Tr}(\boldsymbol{T}^\top \ln(\boldsymbol{T})), \tag{10}$$

where $\ln(\cdot)$ calculates element-wise logarithm of a matrix. The parameter $\tau$ works as the reciprocal of the "temperature" in the smoothed softmax layer in the original distillation method [20, 29].

The principle of our distilled learning method is that when updating $\boldsymbol{B}$ and $\boldsymbol{\Lambda}$, the smoothed underlying distance is used to provide "weak" guidance. Consequently, the student (*i.e.*, the proposed new model with updated $\boldsymbol{B}$ and $\boldsymbol{\Lambda}$) will not completely rely on the information from the teacher (*i.e.*, the underlying distance obtained in a previous iteration), and will tend to explore new basis and weights. In summary, the optimization problem for learning the Wasserstein topic model is

$$\min_{\boldsymbol{A}, \boldsymbol{R}} \mathcal{L}_\tau(\boldsymbol{A}, \boldsymbol{R}) = \min_{\boldsymbol{A}, \boldsymbol{R}} \sum_{m=1}^{M} \mathcal{L}(\boldsymbol{y}_m, \ \boldsymbol{y}_{S_\epsilon}(\boldsymbol{B}(\boldsymbol{R}), \boldsymbol{\lambda}_m(\boldsymbol{A}); \boldsymbol{D}_\theta^\tau)), \tag{11}$$

which can be solved under the same algorithmic framework as that in [36].

Our algorithm is shown in Algorithm 1. The details of the algorithm and the influence of our distilled learning strategy on the convergence of the algorithm are given in the Supplementary Material. Note that our method is compatible with existing techniques, which can work as a fine-tuning method when the underlying distance is initialized by predefined embeddings. When the topic of each document is given, $k_m$ in (6) is predefined and the proposed method can work in a supervised way.

## 4   Related Work

**Word embedding, topic modeling, and their application to clinical data** Traditional topic models, like latent Dirichlet allocation (LDA) [8] and its variants, rely on the "bag-of-words" representation of documents. Word embedding [30] provides another choice, which represents documents as the fusion of the embeddings [27]. Recently, many new word embedding techniques have been proposed, *e.g.*, the Glove in [33] and the linear ensemble embedding in [32], which achieve encouraging

performance on word and document representation. Some works try to combine word embedding and topic modeling. As discussed above, they either use word embeddings as features for topic models [38, 15] or regard topics as labels when learning embeddings [41, 28]. A unified framework for learning topics and word embeddings was still absent prior to this paper.

Focusing on clinical data analysis, word embedding and topic modeling have been applied to many tasks. Considering ICD code assignment as an example, many methods have been proposed to estimate the ICD codes based on clinical records [39, 5, 31, 22], aiming to accelerate diagnoses. Other tasks, like clustering clinical data and the prediction of treatments, can also be achieved by NLP techniques [4, 19, 11].

**Wasserstein learning and its application in NLP** The Wasserstein distance has been proven useful in distribution estimation [9], alignment [44] and clustering [1, 43, 14], avoiding over-smoothed intermediate interpolation results. It can also be used as loss function when learning generative models [12, 3]. The main bottleneck of the application of Wasserstein learning is its high computational complexity. This problem has been greatly eased since Sinkhorn distance was proposed in [13]. Based on Sinkhorn distance, we can apply iterative Bregman projection [7] to approximate Wasserstein distance, and achieve a near-linear time complexity [2]. Many more complicated models have been proposed based on Sinkhorn distance [16, 36]. Focusing on NLP tasks, the methods in [26, 21] use the same framework as ours, computing underlying distances based on word embeddings and measuring the distance between documents in the Wasserstein space. However, the work in [26] does not update the pretrained embeddings, while the model in [21] does not have a hierarchical architecture for topic modeling.

**Model distillation** As a kind of transfer learning techniques, model distillation was originally proposed to learn a simple model (student) under the guidance of a complicated model (teacher) [20]. When learning the target-distilled model, a regularizer based on the smoothed outputs of the complicated model is imposed. Essentially, the distilled complicated model provides the target model with some privileged information [29]. This idea has been widely used in many applications, *e.g.*, textual data modeling [23], healthcare data analysis [10], and image classification [18]. Besides transfer learning, the idea of model distillation has been extended to control the learning process of neural networks [34, 35, 42]. To the best of our knowledge, our work is the first attempt to combine model distillation with Wasserstein learning.

## 5 Experiments

To demonstrate the feasibility and the superiority of our distilled Wasserstein learning (DWL) method, we apply it to analysis of admission records of patients, and compare it with state-of-the-art methods. We consider a subset of the MIMIC-III dataset [25], containing $11,086$ patient admissions, corresponding to $56$ diseases and $25$ procedures, and each admission is represented as a sequence of ICD codes of the diseases and the procedures. Using different methods, we learn the embeddings of the ICD codes and the topics of the admissions and test them on three tasks: mortality prediction, admission-type prediction, and procedure recommendation. For all the methods, we use $50\%$ of the admissions for training, $25\%$ for validation, and the remaining $25\%$ for testing in each task. For our method, the embeddings are obtained by the linear projection of one-hot representations of the ICD codes, which is similar to the Word2Vec [30] and the Doc2Vec [27]. For our method, the loss function $\mathcal{L}$ is squared loss. The hyperparameters of our method are set via cross validation: the batch size $s = 256$, $\beta = 0.01$, $\epsilon = 0.01$, the number of topics $K = 8$, the embedding dimension $D = 50$, and the learning rate $\rho = 0.05$. The number of epochs $I$ is set to be $5$ when the embeddings are initialized by Word2Vec, and $50$ when training from scratch. The distillation parameter is $\tau = 0.5$ empirically, whose influence on learning result is shown in the Supplementary Material.

### 5.1 Admission classification and procedure recommendation

The admissions of patients often have a clustering structure. According to the seriousness of the admissions, they are categorized into four classes in the MIMIC-III dataset: *elective*, *emergency*, *urgent* and *newborn*. Additionally, diseases and procedures may lead to mortality, and the admissions can be clustered based on whether the patients die or not during their admissions. Even if learned in a unsupervised way, the proposed embeddings should reflect the clustering structure of the admissions to some degree. We test our DWL method on the prediction of admission type and

Table 1: Admission classification accuracy (%) for various methods.

| Word Feature | Doc. Feature | Metric | Dim. | Mortality | | Adm. Type | |
|---|---|---|---|---|---|---|---|
| | | | | 1-NN | 5-NN | 1-NN | 5-NN |
| — | TF-IDF [17] | | 81 | 69.98±0.05 | 75.32±0.04 | 82.27±0.03 | 88.28±0.02 |
| — | LDA [8] | | 8 | 66.03±0.06 | 69.05±0.06 | 81.41±0.04 | 86.57±0.04 |
| Word2Vec [30] | Doc2Vec [27] | | 50 | 57.98±0.08 | 59.80±0.08 | 70.57±0.08 | 79.94±0.07 |
| Word2Vec [30] | AvePooling | Euclidean | 50 | 70.42±0.05 | 75.21±0.04 | 84.88±0.07 | 89.16±0.06 |
| Glove [33] | AvePooling | | 50 | 66.94±0.06 | 73.21±0.04 | 81.91±0.05 | 88.21±0.05 |
| DWL (Scratch) | AvePooling | | 50 | 71.01±0.12 | 74.74±0.11 | 84.54±0.13 | **89.49±0.12** |
| DWL (Finetune) | AvePooling | | 50 | **71.52±0.07** | **75.44±0.07** | **85.54±0.09** | 89.28±0.09 |
| Word2Vec [30] | | | | 70.31±0.04 | 74.89±0.04 | 83.63±0.05 | **89.25±0.04** |
| DWL (Scratch) | Topic weight [36] | Euclidean | 8 | 70.45±0.08 | 74.88±0.07 | 83.82±0.12 | 88.80±0.12 |
| DWL (Finetune) | | | | **70.88±0.07** | **75.67±0.07** | **84.26±0.09** | 89.13±0.08 |
| Word2Vec [30] | | | | 70.61±0.04 | 75.92±0.04 | 84.08±0.05 | 89.06±0.05 |
| Glove [33] | Word distribution | Wasserstein [26] | 81 | 70.64±0.06 | 75.97±0.05 | 83.92±0.08 | 89.17±0.07 |
| DWL (Scratch) | | | | **71.01±0.10** | 75.88±0.09 | 84.23±0.12 | 89.33±0.11 |
| DWL (Finetune) | | | | 70.65±0.07 | **76.00±0.06** | **84.35±0.08** | **89.61±0.07** |

Table 2: Top-$N$ procedure recommendation results for various methods.

| Method | Top-1 (%) | | | Top-3 (%) | | | Top-5 (%) | | |
|---|---|---|---|---|---|---|---|---|---|
| | P | R | F1 | P | R | F1 | P | R | F1 |
| Word2Vec [30] | 39.95 | 13.27 | 18.25 | 31.70 | 33.46 | 29.30 | 28.89 | 46.98 | 32.59 |
| Glove [33] | 32.66 | 13.01 | 17.22 | 29.45 | 30.99 | 27.41 | 27.93 | 44.79 | 31.47 |
| DWL (Scratch) | 37.89 | 12.42 | 17.16 | 30.14 | 29.78 | 27.14 | 27.39 | 43.81 | 30.81 |
| DWL (Finetune) | **40.00** | **13.76** | **18.71** | **31.88** | **33.71** | **29.58** | **30.59** | **48.56** | **34.28** |

mortality. For the admissions, we can either represent them by the distributions of the codes and calculate the Wasserstein distance between them, or represent them by the average pooling of the code embeddings and calculate the Euclidean distance between them. A simple KNN classifier can be applied under these two metrics, and we consider $K = 1$ and $K = 5$. We compare the proposed method with the following baselines: ($i$) bag-of-words-based methods like TF-IDF [17] and LDA [8]; ($ii$) word/document embedding methods like Word2Vec [30], Glove [33], and Doc2Vec [27]; and ($iii$) the Wasserstein-distance-based method in [26]. We tested various methods in 20 trials. In each trial, we trained different models on a subset of training admissions and tested them on the same testing set, and calculated the averaged results and their $90\%$ confidential intervals.

The classification accuracy for various methods are shown in Table 1. Our DWL method is superior to its competitors on classification accuracy. Besides this encouraging result, we also observe two interesting and important phenomena. First, for our DWL method the model trained from scratch has comparable performance to that fine-tuned from Word2Vec's embeddings, which means that our method is robust to initialization when exploring clustering structure of admissions. Second, compared with measuring Wasserstein distance between documents, representing the documents by the average pooling of embeddings and measuring their Euclidean distance obtains comparable results. Considering the fact that measuring Euclidean distance has much lower complexity than measuring Wasserstein distance, this phenomenon implies that although our DWL method is time-consuming in the training phase, the trained models can be easily deployed for large-scale data in the testing phase.

The third task is recommending procedures according to the diseases in the admissions. In our framework, this task can be solved by establishing a bipartite graph between diseases and procedures based on the Euclidean distance between their embeddings. The proposed embeddings should reflect the clinical relationships between procedures and diseases, such that the procedures are assigned to the diseases with short distance. For the $m$-th admission, we may recommend a list of procedures with length $L$, denoted as $E_m$, based on its diseases and evaluate recommendation results based on the ground truth list of procedures, denoted as $T_m$. In particular, given $\{E_m, T_m\}$, we calculate the top-$L$ precision, recall and F1-score as follows: $P = \sum_{m=1}^{M} P_m = \sum_{m=1}^{M} \frac{|E_m \cap T_m|}{|E_m|}$, $R = \sum_{m=1}^{M} R_m = \sum_{m=1}^{M} \frac{|E_m \cap T_m|}{|T_m|}$, $F1 = \sum_{m=1}^{M} \frac{2P_m R_m}{P_m + R_m}$. Table 2 shows the performance of various methods with $L = 1, 3, 5$. We find that although our DWL method is not as good as the Word2Vec when the model is trained from scratch, which may be caused by the much fewer epochs we executed, it indeed outperforms other methods when the model is fine-tuned from Word2Vec.

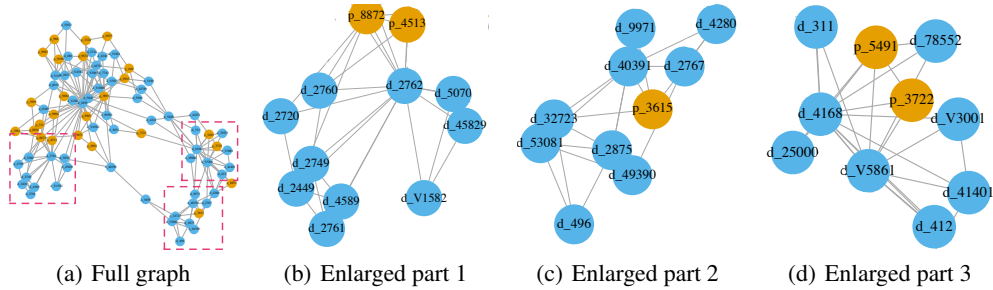

| (a) Full graph | (b) Enlarged part 1 | (c) Enlarged part 2 | (d) Enlarged part 3 |

Figure 2: (a) The KNN graph of diseases and procedures with $K = 4$. Its enlarged version is in the Supplementary Material. The ICD codes related to diseases are with a prefix "d", whose nodes are blue, while those related to procedures are with a prefix "p", whose nodes are orange. (b-d) Three enlarged subgraphs corresponding to the red frames in (a). In each subfigure, the nodes/dots in blue are diseases while the nodes/dots in orange are procedures.

Table 3: Top-3 ICD codes in each topic associated with the corresponding diseases/procedures.

| Topic 1 | Topic 2 | Topic 3 | Topic 4 | Topic 5 | Topic 6 | Topic 7 | Topic 8 |
|---|---|---|---|---|---|---|---|
| d_5859 | d_4241 | d_311 | p_8856 | d_2449 | d_7742 | p_9904 | d_311 |
| Chronic kidney disease | Aortic valve disorders | Mycobacteria | Coronary arteriography | Hypothyroidism | Neonatal jaundice | Cell transfusion | Mycobacteria |
| d_2859 | p_3891 | d_V3001 | d_41071 | d_2749 | p_9672 | d_5119 | d_5119 |
| Anemia | Arterial catheterization | Single liveborn | Subendocardial infarction | Gout | Ventilation | Pleural effusion | Pleural effusion |
| p_8872 | d_9971 | d_5849 | d_2851 | d_41401 | p_9907 | p_331 | d_42731 |
| Heart ultrasound | Cardiac complications | Kidney failure | Posthemorrhagic anemia | Atherosclerosis | Serum transfusion | Incision of lung | Atrial fibrillation |

## 5.2 Rationality Analysis

To verify the rationality of our learning result, in Fig. 2 we visualize the KNN graph of diseases and procedures. We can find that the diseases in Fig. 2(a) have obvious clustering structure while the procedures are dispersed according to their connections with matched diseases. Furthermore, the three typical subgraphs in Fig. 2 can be interpreted from a clinical viewpoint. Figure 2(b) clusters cardiovascular diseases like hypotension (d_4589, d_45829) and hyperosmolality (d_2762) with their common procedure, $i.e.$, diagnostic ultrasound of heart (p_8872). Figure 2(c) clusters coronary artery bypass (p_3615) with typical postoperative responses like hyperpotassemia (d_2767), cardiac complications (d_9971) and congestive heart failure (d_4280). Figure 2(d) clusters chronic pulmonary heart diseases (d_4168) with its common procedures like cardiac catheterization (p_3772) and abdominal drainage (p_5491) and the procedures are connected with potential complications like septic shock (d_78552). The rationality of our learning result can also be demonstrated by the topics shown in Table 3. According to the top-3 ICD codes, some topics have obvious clinical interpretations. Specifically, topic 1 is about kidney disease and its complications and procedures; topic 2 and 5 are about serious cardiovascular diseases; topic 4 is about diabetes and its cardiovascular complications and procedures; topic 6 is about the diseases and the procedures of neonatal. We show the map between ICD codes and corresponding diseases/procedures in the Supplementary Material.

## 6 Conclusion and Future Work

We have proposed a novel method to jointly learn the Euclidean word embeddings and a Wasserstein topic model in a unified framework. An alternating optimization method was applied to iteratively update topics, their weights, and the embeddings of words. We introduced a simple but effective model distillation method to improve the performance of the learning algorithm. Testing on clinical admission records, our method shows the superiority over other competitive models for various tasks. Currently, the proposed learning method shows a potential for more-traditional textual data analysis (documents), but its computational complexity is still too high for large-scale document applications (because the vocabulary for real documents is typically much larger than the number of ICD codes considered here in the motivating hospital-admissions application). In the future, we plan to further accelerate the learning method, $e.g.$, by replacing the Sinkhorn-based updating precedure with its variants like the Greenkhorn-based updating method [2].

# 7 Acknowledgments

This research was supported in part by DARPA, DOE, NIH, ONR and NSF. Morgan A. Schmitz kindly helped us by sharing his Wasserstein dictionary learning code. We also thank Prof. Hongyuan Zha at Georgia Institute of Technology for helpful discussions.

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
