[Supplementary Material · dwl_cr_supplementary.pdf]

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

# Appendix

## The derivation of Sinkhorn gradient

The key part of our learning algorithm is calculating Sinkhorn gradient given the distilled underlying distance matrix $\boldsymbol{D}_\theta^\tau$. Same to the method in [36], we use the following algorithm to calculate $\nabla_{\boldsymbol{B}}\mathcal{L}_\tau$ and $\nabla_{\boldsymbol{\lambda}_m}\mathcal{L}_\tau$ for each document $\boldsymbol{y}_m$. Here, $\odot$ is element-wise multiplication, $\dot{\div}$ is element-wise

---

**Algorithm 2** Computation of Sinkhorn gradient

---

1: **Input:** Arbitrary document $\boldsymbol{y}$. Underlying distance $\boldsymbol{D}_\theta^\tau$. Distillation parameter $\tau$. The number of inner iteration $L$. The weight in Sinkhon distance $\epsilon$. Current basis $\boldsymbol{B} = [\boldsymbol{b}_k]$ and weights $\boldsymbol{\lambda} = [\lambda_k]$.

2: **Output:** $\nabla_{\boldsymbol{B}}\mathcal{L}_\tau$ and $\nabla_{\boldsymbol{\lambda}}\mathcal{L}_\tau$.

3: Calculate $\boldsymbol{C} = \exp(\frac{\boldsymbol{D}_\theta^\tau}{\epsilon})$.

4: **Forward loop:**

5: Initialize $\boldsymbol{\beta}_k^0 = \mathbf{1}_N$ for $k = 1, ..., K$.

6: **for** $l = 1, ..., L$ **do**

7:    $\boldsymbol{\phi}_k^l = \boldsymbol{C}^\top \frac{\boldsymbol{b}_k}{\boldsymbol{C}\boldsymbol{\beta}_k^{l-1}}$ for $k = 1, ..., K$.

8:    $\hat{\boldsymbol{y}} = \Pi_k(\boldsymbol{\phi}_k^l)^{\lambda_k}$.

9:    $\boldsymbol{\beta}_k^l = \frac{\hat{\boldsymbol{y}}}{\boldsymbol{\phi}_k^l}$.

10: **end for**

11: **Backward loop for weights:**

12: Initialize $\boldsymbol{w} = [w_k] = \mathbf{0}_K, \boldsymbol{r} = [\boldsymbol{r}_k] = \mathbf{0}_{N \times K}, \boldsymbol{g} = \nabla\mathcal{L}(\hat{\boldsymbol{y}}, \boldsymbol{y}) \odot \hat{\boldsymbol{y}}$.

13: **for** $l = 1, ..., L$ **do**

14:    $w_k = w_k + (\ln\boldsymbol{\phi}_k^l)^\top \boldsymbol{g}$ for $k = 1, ..., K$.

15:    $\boldsymbol{r}_k = -\boldsymbol{C}^\top \left( \boldsymbol{C} \left( \frac{\lambda_k \boldsymbol{g} - \boldsymbol{r}_k}{\boldsymbol{\phi}_k^l} \right) \odot \frac{\boldsymbol{b}_k}{(\boldsymbol{C}\boldsymbol{\beta}_k^{l-1})^2} \right) \odot \boldsymbol{\beta}_k^{l-1}$ for $k = 1, ..., K$.

16:    $\boldsymbol{g} = \sum_k \boldsymbol{r}_k$

17: **end for**

18: **Backward loop for basis:**

19: Initialize $\boldsymbol{M} = [\boldsymbol{m}_k] = \mathbf{0}_{N \times K}, \boldsymbol{Z} = [\boldsymbol{z}_k] = \mathbf{0}_{N \times K}$.

20: **for** $l = 1, ..., L$ **do**

21:    $\boldsymbol{\psi}_k = \boldsymbol{C}((\lambda_k \nabla\mathcal{L}(\hat{\boldsymbol{y}}, \boldsymbol{y}) - \boldsymbol{z}_k) \odot \boldsymbol{\beta}_k^l)$.

22:    $\boldsymbol{m}_k = \boldsymbol{m}_k + \frac{\boldsymbol{\psi}_k}{\boldsymbol{C}\boldsymbol{\beta}_k^{l-1}}$.

23:    $\boldsymbol{z}_k = -\frac{\mathbf{1}_N}{\boldsymbol{\phi}_k^{l-1}} \odot \boldsymbol{C}^\top \frac{\boldsymbol{b}_k \odot \boldsymbol{\psi}_k}{(\boldsymbol{C}\boldsymbol{\beta}_k^{l-1})^2}$.

24:    $\nabla\mathcal{L}(\hat{\boldsymbol{y}}, \boldsymbol{y}) = \sum_k \boldsymbol{z}_k$.

25: **end for**

26: $\nabla_{\boldsymbol{B}}\mathcal{L}_\tau = \boldsymbol{M}$ and $\nabla_{\boldsymbol{\lambda}}\mathcal{L}_\tau = \boldsymbol{w}$.

---

division, $(\cdot)^2$ is element-wise square, and $\ln(\cdot)$ is element-wise logarithm. More details of the algorithm can be found in [36].

## Influence of distillation parameters

The distillation parameter $\tau$ has significant influence on the convergence and the performance of our learning algorithm. We visualize the convergence rate of our DWL method with respect to different $\tau$'s in the task of admission type prediction. In Fig. 3, we can find that when $\tau = 1$, which means that the model is learned without distillation, the increase of training accuracy is very slow because of the gradient vanishment problem. On the contrary, when $\tau = 0.25$, which means that we use model distillation heavily in the training phase and the "student" leverages little information from "teacher", the training accuracy increases rapidly but converges to an unsatisfying level. This is because the distilled underlying distance is over-smoothed, which cannot provide sufficient guidance to further update basis and weights. To achieve a trade-off between the convergence and the performance of our algorithm, finally we choose $\tau = 0.5$ empirically according to the experimental results.

It should be noted that although we set the distillation parameter empirically, as [20, 29] did, we give a reasonable range: $\tau$ should be smaller than 1 (to achieve distillation) and larger than 0.25 (to avoid oversmoothness). We will study the setting of the parameter in our future work.

Figure 3: The convergence of our DWL method with respect to $\tau$'s in the task of admission type prediction.

**Sentiment analysis on Twitter dataset**

Besides the MIMIC-III dataset, we compared our method against the Wasserstein-distance based method [26] on sentiment analysis based on the Twitter dataset in that paper. Our method obtains comparable results, i.e., $28.92 \pm 0.14\%$ testing error, which is slightly lower than that in [26].

**The enlarged graph of ICD codes**

The Fig. 2(a) in the paper is enlarged and shown below for better visual effect. The map between ICD codes and diseases/procedures is attached as well.

Figure 4: The enlarged KNN graph of diseases and procedures with $K = 4$.

Table 4: The map between ICD codes and diseases/procedures

| ICD code | Disease/Procedure |
|---|---|
| d_4019 | Unspecified essential hypertension |
| d_41401 | Coronary atherosclerosis of native coronary artery |
| d_4241 | Aortic valve disorders |
| d_V4582 | Percutaneous transluminal coronary angioplasty status |
| d_2724 | Other and unspecified hyperlipidemia |
| d_486 | Pneumonia, organism unspecified |
| d_99592 | Severe sepsis |
| d_51881 | Acute respiratory failure |
| d_5990 | Urinary tract infection, site not specified |
| d_5849 | Acute kidney failure, unspecified |
| d_78552 | Septic shock |
| d_25000 | Diabetes mellitus without mention of complication, type II or unspecified type |
| d_2449 | Unspecified acquired hypothyroidism |
| d_41071 | Subendocardial infarction, initial episode of care |
| d_4280 | Congestive heart failure, unspecified |
| d_4168 | Other chronic pulmonary heart diseases |
| d_412 | Pneumococcus infection in conditions classified elsewhere and of unspecified site |
| d_2761 | Hyposmolality and/or hyponatremia |
| d_2720 | Pure hypercholesterolemia |
| d_2762 | Acidosis |
| d_389 | Unspecified septicemia |
| d_4589 | Hypotension, unspecified |
| d_42731 | Atrial fibrillation |
| d_2859 | Anemia, unspecified |
| d_311 | Cutaneous diseases due to other mycobacteria |
| d_V3001 | Single liveborn, born in hospital, delivered by cesarean section |
| d_V053 | Need for prophylactic vaccination and inoculation against viral hepatitis |
| d_4240 | Mitral valve disorders |
| d_V3000 | Single liveborn, born in hospital, delivered without mention of cesarean section |
| d_7742 | Neonatal jaundice associated with preterm delivery |
| d_42789 | Other specified cardiac dysrhythmias |
| d_5070 | Pneumonitis due to inhalation of food or vomitus |
| d_V502 | Routine or ritual circumcision |
| d_2760 | Hyperosmolality and/or hypernatremia |
| d_V1582 | Personal history of tobacco use |
| d_40390 | Hypertensive chronic kidney disease, unspecified, with chronic kidney disease stage I through stage IV, or unspecified |
| d_V4581 | Aortocoronary bypass status |
| d_V290 | Observation for suspected infectious condition |
| d_5845 | Acute kidney failure with lesion of tubular necrosis |
| d_2875 | Thrombocytopenia, unspecified |
| d_2767 | Hyperpotassemia |
| d_32723 | Obstructive sleep apnea (adult)(pediatric) |
| d_V5861 | Long-term (current) use of anticoagulants |
| d_2851 | Acute posthemorrhagic anemia |
| d_53081 | Esophageal reflux |
| d_496 | Chronic airway obstruction, not elsewhere classified |
| d_40391 | Hypertensive chronic kidney disease, unspecified, with chronic kidney disease stage V or end stage renal disease |
| d_9971 | Gross hematuria |
| d_5119 | Unspecified pleural effusion |
| d_2749 | Gout, unspecified |
| d_5859 | Chronic kidney disease, unspecified |
| d_49390 | Asthma, unspecified type, unspecified |
| d_45829 | Other iatrogenic hypotension |
| d_3051 | Tobacco use disorder |
| d_V5867 | Long-term (current) use of insulin |
| d_5180 | Pulmonary collapse |
| p_9604 | Insertion of endotracheal tube |
| p_9671 | Continuous invasive mechanical ventilation for less than 96 consecutive hours |
| p_3615 | Single internal mammary-coronary artery bypass |
| p_3961 | Extracorporeal circulation auxiliary to open heart surgery |
| p_8872 | Diagnostic ultrasound of heart |
| p_9904 | Transfusion of packed cells |
| p_9907 | Transfusion of other serum |
| p_9672 | Continuous invasive mechanical ventilation for 96 consecutive hours or more |
| p_331 | Spinal tap |
| p_3893 | Venous catheterization, not elsewhere classified |
| p_966 | Enteral infusion of concentrated nutritional substances |
| p_3995 | Hemodialysis |
| p_9915 | Parenteral infusion of concentrated nutritional substances |
| p_8856 | Coronary arteriography using two catheters |
| p_9955 | Prophylactic administration of vaccine against other diseases |
| p_3891 | Arterial catheterization |
| p_9390 | Non-invasive mechanical ventilation |
| p_9983 | Other phototherapy |
| p_640 | Circumcision |
| p_3722 | Left heart cardiac catheterization |
| p_8853 | Angiocardiography of left heart structures |
| p_3723 | Combined right and left heart cardiac catheterization |
| p_5491 | Percutaneous abdominal drainage |
| p_3324 | Closed (endoscopic) biopsy of bronchus |
| p_4513 | Other endoscopy of small intestine |