[Reviews · NeurIPS 2018]

Reviewer 1



Summary ======= The authors present a Distilled Wasserstein Learning (DWL) method for simultaneously learning a topic model alongside a word embedding using a model/approach based on the Wasserstein distance applied to elements of finite simplices. This is claimed as the first such method to simultaneously fit topics alongside embeddings. In particular, their embeddings only exploit on document co-occurence rather than nearby co-occurence within a sequence (i.e. using word order information) such as with word2vec. The authors demonstrate the superiority of their embeddings against a variety of benchmarks on three tasks: mortality prediction, admissions-type prediction, and procedure recommendation, using a single corpus of patient admission records where words are the international classification of diseases (ICD) ids of procedures and diseases. There are a number of apparently novel features to their approach which they outline in their paper, namely: * It is a topic model where observed word frequencies within a document are approximated as the *barycentres* (centre of mass) of a weighted sum over a low rank basis of topics (where these barycentres are with respect to some Wasserstein distance). This is a compelling idea, as the Wasserstein distance is a very natural one for probability distributions. * Development of an objective function, allowing alternating optimisation. One phase learning the documents as distributions of topics (the weights \Lambda) and the topics as distributions of words (the basis B). The other phase refining the word embeddings. * A distilling approach to avoid vanishing gradients when learning the weights \Lambda and basis B. This looks like a regularised version of the objective, which takes advantage of Sinkhorn distance and a fractional powers of the underlying embedding distance. * A smoothness regulariser on the parameters, \theta, of the embedding function, meaning it is not far from the previous iteration's value. * It is unclear whether this method will scale. The paper is well written, and the method appears reproducible. Further, it is both dense in mathematical justification, and clearly references related work and foundational results. To my knowledge it is novel and represents a significant contribution which would be of interest to the field. I have a few concerns about the following aspects of the paper: * There is no independent verification/clear justification that word order in the patient record data is less consistent than in natural language corpora. We rely on the superiority of this model over those that utilise word order information for their embeddings. * Sometimes the intuitive explanations fall a little short. * There is a heavy reliance throughout on foundational results from an arXiv paper [36] (hence not peer reviewed). * For the regularisation/distilling approaches, there don't appear to be any theoretical results assessing their influence on the convergence/optimality of the overall model. Moreover, they involve choosing hyperparameters whose influence it is difficult to assess other than empirically, * The evaluation is done on just one data-set (with three related tasks), and while the results show that the proposed method consistently outperforms other existing approaches the improvements are modest and there are no errorbars/significance results. Nonetheless, I am recommending the paper for acceptance, as I feel the positive aspects outweight these negatives. Detail ====== ### p1, lines 31-33 > ICD codes for a given patient admission is less relevant, as there is often a diversity in the observed codes for a given admission, and code order may hold less meaning This could be more convincing. There is diversity in the words that appear around other words in natural language. Could you verify this on data? ### p2, figure 1 I think it is possibly trying to do a little too much in one figure, or it might be the choice of perspective. Having read the paper from beginning to end, it is clearer, but it would have helped having the associated intuition earlier. ### p3, lines 92-93 > Π(u, v) is the set of all probability measures on Ω × Ω. I think there should be an addional requirement that the left and right marginals are u and v respectively, e.g. given some $$\gamma \in \Pi(u,v)$$ then $$\gamma(\Omega,A) = v(A) \quad \text{for all } A \subseteq \Omega$$ ### p3-p4, use of the word transfer > ...the transfer between different topics (and different documents) is achieved by the optimal transport built on the embedding-based underlying distance. For arbitrary two word embeddings, the more similar they are, the smaller ground distance we have, and more easily we can achieve transfer between them. This is a little ambiguous and could be made clearer. ### p7, table 1 Although the performance is consistently better the results only represent a seemingly modest improvement (especially considering the increased computational complexity). ### p7, line 263 The variables P_m and R_m are not defined (although at a guess they are the summands of the precision and recall respectively. ### p8, line 270, clarity > dispersed mainly accordingly to their connections with matched diseases. Not sure what is meant by mainly (or how that is determined). How is this to be evaluated. It is unclear how anyone other than a medical expert would be able to evaluate how meaningful this clustering is, without some additional information (even given the additional information in the supplementary materials).

Reviewer 2



This paper proposed a new framework to jointly learn "word" embedding and "topical" mixture of documents. The proposed approach is based on Wasserstein topic model build on the word-embedding space. The proposed approach was applied to several tasks related to medical records data. The paper is overall solid and well organized. I have a few concerns regarding the experiment validation - only very simple baselines (word2vec like embeddings, or LDA) are demonstrated. As author pointed out in related work, there have been recently seleval attempts in applying NLP tools for very similar dataset/tasks. There are also other attempts to combine LDA and Word2vec in the same framework. Considering a few based lines in each of these aspects would be much more convincing. - since are generative model were proposed, some analysis/metrics like likelihood and per-plexity would be good to demonstrate how the proposed model fits the real-world observations - it would be good to consider a few different datasets in this domain to demonstrate the descriptive power of the proposed model. (e.g., in Citation[11])

Reviewer 3



The main idea: this paper proposes a novel Wasserstein method based on the underlying word (code) embedding. The Wasserstein method can measure the distance between topics. The topic word distributions, their optimal transport to the word distributions of documents, and word embeddings are jointly learnt. The paper also proposes a distilled ground-distance matrix when updating the topic distributions and smoothly calculate the optimal transports. The method achieves strong performance on the task of mortality prediction,admission-type prediction, and procedure recommendation. Compared with prior work, this paper proposes a unified framework that could jointly learn topics and word embeddings. Also, this work updates the pretrained embeddings rather than freeze them, and have a hierarchical architecture for topic modeling. No prior work has combined Wasserstein learning with knowledge distillation. Strength: this paper is fairly well written and the idea is very clear. The method is novel in that it proposes a Wasserstein topic model based on word embedding. It jointly learns topic distribution and word embeddings. And it incorporates knowledge distillation in the update of topic models. The experimental results show improvement over baseline models. Weakness: It'd be great if the author could be more explicit on the motivation of using Wasserstein method to model distance between admissions. It's not that straightforward from the current writing. The author only describes the limitation of previous methods. The author mentions that their method outperforms state-of-art methods. It's unclear to me which method in table 1 is the state-of-art. Word2vec/Glove and AvePooling and Doc2Vec seems a bit weak baseline. Related to the above question, have you tried other classifier other than KNN? For example, it could a simple neural network that fine-tunes the pre-trained word2vec/Glove. This could be a better baseline than using fixed pre-trained embedding in KNN.